Seasonal and successional dynamics of size-dependent plant demographic rates in a tropical dry forest

Saenz-Pedroza Irving 1
Feldman Richard 1
Reyes-García Casandra 1
Meave Jorge A. 2
Calvo-Irabien Luz Maria 1
May-Pat Filogonio 1
Dupuy Juan M. jmdupuy@cicy.mx 1
1 Unidad de Recursos Naturales, Centro de Investigación Científica de Yucatán , Mérida , Yucatán , México
2 Departamento de Ecología y Recursos Naturales, Facultad de Ciencias, Universidad Nacional Autónoma de México , Coyoacán , Ciudad de México , México
Sosa Victoria
Electronic publication date: 2020 Sep 14
Publication date: 2020
Volume: 8
Electronic Location ID: e9636
Received 2020 Jan 10; Accepted 2020 Jul 9
Copyright: ©2020 Saenz-Pedroza et al.
Copyright year: 2020
Copyright holder: Saenz-Pedroza et al.
License: This is an open access article distributed under the terms of the Creative Commons Attribution License, which permits unrestricted use, distribution, reproduction and adaptation in any medium and for any purpose provided that it is properly attributed. For attribution, the original author(s), title, publication source (PeerJ) and either DOI or URL of the article must be cited.
License URL: https://creativecommons.org/licenses/by/4.0/

Keywords: Chronosequence resampling, Plant and species density, Recruitment, Mortality, Dry season, Rainy season, Secondary forest succession, Competition, Environmental filtering

Funding: Consejo Nacional de Ciencia y Tecnología (CONACYT) 265363 Consejo Nacional de Ciencia y Tecnología (CONACYT) and Secretaría de Medio Ambiente y Recursos Naturales (SEMARNAT) [Fondo Sectorial de Investigación Ambiental 2004-C01-227 to Juan Manuel Dupuy] Consejo Nacional de Ciencia y Tecnología (CONACYT) and Government of the State of Yucatán [Fondo Mixto YUC2004-003-027 to Juan Manuel Dupuy] This work was supported by the Consejo Nacional de Ciencia y Tecnología (CONACYT), [265363 to Irving Saenz], jointly with the Secretaría de Medio Ambiente y Recursos Naturales (SEMARNAT) [Fondo Sectorial de Investigación Ambiental 2004-C01-227 to Juan Manuel Dupuy], and the Government of the State of Yucatán [Fondo Mixto YUC2004-003-027 to Juan Manuel Dupuy]. There was no additional external funding received for this study. The funders had no role in study design, data collection and analysis, decision to publish, or preparation of the manuscript.

==============================
Tropical forests are globally important for biodiversity conservation and climate change mitigation but are being converted to other land uses. Conversion of seasonally dry tropical forests (SDTF) is particularly high while their protection is low. Secondary succession allows forests to recover their structure, diversity and composition after conversion and subsequent abandonment and is influenced by demographic rates of the constituent species. However, how these rates vary between seasons for different plant sizes at different successional stages in SDTF is not known. The effect of seasonal drought may be more severe early in succession, when temperature and radiation are high, while competition and density-dependent processes may be more important at later stages, when vegetation is tall and dense. Besides, the effects of seasonality and successional stage may vary with plant size. Large plants can better compete with small plants for limiting resources and may also have a greater capacity to withstand stress. We asked how size-dependent density, species density, recruitment and mortality varied between seasons and successional stages in a SDTF. We monitored a chronosequence in Yucatan, Mexico, over six years in three 0.1 ha plots in each of three successional stages: early (3–5 years-old), intermediate (18–20 years-old) and advanced (>50 years-old). Recruitment, mortality and species gain and loss rates were calculated from wet and dry season censuses separately for large (diameter > 5 cm) and small (1–5 cm in diameter) plants. We used linear mixed-effects models to assess the effects of successional stage, seasonality and their changes through time on demographic rates and on plant and species density. Seasonality affected demographic rates and density of large plants, which exhibited high wet-season recruitment and species gain rates at the early stage and high wet-season mortality at the intermediate stage, resulting in an increase in plant and species density early in succession followed by a subsequent stabilization. Small plant density decreased steadily after only 5 years of land abandonment, whereas species density increased with successional stage. A decline in species dominance may be responsible for these contrasting patterns. Seasonality, successional stage and their changes through time had a stronger influence on large plants, likely because of large among-plot variation of small plants. Notwithstanding the short duration of our study, our results suggest that climate-change driven decreases in rainy season precipitation may have an influence on successional dynamics in our study forest as strong as, or even stronger than, prolonged or severe droughts during the dry season.

Introduction

Tropical forests are globally important reservoirs of biodiversity and play a major role in the global carbon cycle (Pan et al., 2011; Slik et al., 2015), but are being converted to agricultural and pasturelands at alarming rates (FAO, 2015). Conversion of seasonally dry tropical forests (SDTF) is particularly high, while their protection is low, making them one of the most threatened ecosystems worldwide (Janzen, 1988; Miles et al., 2006; Aide et al., 2013). At present, this biome consists mainly of secondary vegetation re-growing after converted land is abandoned (Sánchez-Azofeifa & Portillo-Quintero, 2011). Understanding tropical forest succession is therefore essential to elucidate the role of secondary forests in biodiversity conservation (Derroire et al., 2016; Rozendaal et al., 2019), carbon dynamics and climate change mitigation (Saatchi et al., 2011; Becknell, Kissing Kucek & Powers, 2012; Chazdon et al., 2016). How the structure, diversity and composition of regenerating communities change during secondary succession is influenced by the aggregated demographic rates of the constituent species: recruitment, growth and mortality (Van Breugel, Martínez-Ramos & Bongers, 2006; Rozendaal & Chazdon, 2015; Muscarella et al., 2017; Rozendaal et al., 2017). However, given the dominance of annual surveys in the study of SDTF succession, it has not been possible to assess how these demographic rates are influenced by seasonality and if the effects of seasonality vary along succession and with plant size. Climate change may affect the duration and severity of dry-season drought, and the amount of rainy season precipitation (Allen et al., 2017), emphasizing the need to understand the effects of seasonality on plant demographic rates along SDTF succession.

Generally, tropical forests can recover their structure and diversity within a few decades after land abandonment through secondary succession (Guariguata & Ostertag, 2001; Chazdon, 2014; Poorter et al., 2016), although recovery of their species composition may take centuries (Rozendaal et al., 2019). In particular, plant density can recover quickly but at rates that vary tremendously among individual forest stands (Aide et al., 2000; Kennard, 2002; Chazdon et al., 2007; Lebrija-Trejos et al., 2008). The fast recovery of plant density is strongly associated with high recruitment rates resulting from a combination of resprouting of remnant vegetation and germination of seeds either stored in the soil seedbank or dispersed into the site (seed rain) after agricultural lands are abandoned (Chazdon, 2014). Species density (the number of species per sampled area) also increases over succession in tropical forests (Guariguata & Ostertag, 2001; Chazdon et al., 2007). Although species density is tightly linked to plant density (Denslow, 1995; Guariguata & Ostertag, 2001), strong dominance by one or few species can lead to a decoupling of these two forest attributes (Mesquita et al., 2001; Dupuy et al., 2012). Strong dominance can also influence community-level demographic rates—growth, recruitment and mortality—in tropical forests (Van Breugel, Martínez-Ramos & Bongers, 2006).

Rainfall seasonality interacts with environmental gradients underlying SDTF succession. As succession proceeds and aboveground biomass recovers, the abiotic environment shifts from water-stressed conditions (i.e., high temperature, solar radiation and water vapor-pressure deficit, and low soil water content) in recently abandoned stands to moister, cooler and shadier environments in older stands (Lebrija-Trejos et al., 2011; Pineda-García, Paz & Meinzer, 2013). Thus, the harsh conditions that prevail early in succession may be further exacerbated during the dry season and represent an even stronger filter to plant survival, compared to the more mesic conditions of late-successional stages, which may somewhat buffer water stress during the dry season (McLaren & McDonald, 2003; Lebrija-Trejos et al., 2010b; Maza-Villalobos, Poorter & Martínez-Ramos, 2013). Seasonality delineates temporal patterns of carbon assimilation in SDTF constraining growth and recruitment mostly to the rainy season, whereas drought during the dry season can strongly reduce plant establishment and growth and increase mortality risk (Murphy & Lugo, 1986; Khurana & Singh, 2001; Ceccon, Huante & Rincón, 2006; Becknell, Kissing Kucek & Powers, 2012; Alberton et al., 2019; Campos et al., 2019) especially early in succession. Although seasonality affects SDTF regeneration and secondary succession processes, few studies have examined the effects of seasonality on plant community dynamics during SDTF succession (Lebrija-Trejos et al., 2011; Maza-Villalobos, Poorter & Martínez-Ramos, 2013).

Plant successional patterns and the underlying demographic rates can vary with plant size. Successional patterns of plant and species density are known to be size dependent, with smaller plants recovering earlier in succession and showing greater variation among forest stands than larger plants (Guariguata & Ostertag, 2001; Kennard, 2002; Dupuy et al., 2012). This may be partly due to strong dominance in small plants and a decrease in dominance and increase in diversity with increasing plant size, as a result of density-dependent processes and community compensatory trends operating on plant ontogeny (Webb & Peart, 1999; Harms et al., 2000). Moreover, large plants have a more developed root and shoot system, which allows them greater access to resources and confers them greater capacity to store carbon and water than small plants (Schwinning & Weiner, 1998; Niinemets, 2010; Ma et al., 2016). Therefore, size-asymmetric competition favors large-size plants, resulting in higher growth and lower mortality rates compared to small plants (Van Breugel, Martínez-Ramos & Bongers, 2006). Large plants may also have a greater capacity to withstand biotic and abiotic stress than small plants, although, at the higher end of the plant size spectrum, a recent study in an Amazonian SDTF found that large trees are actually more vulnerable to xylem embolism and drought stress, and may suffer greater drought-related mortality than smaller ones (Brum et al., 2019). Seasonal variation in microenvironmental conditions (Lebrija-Trejos et al., 2011; Méndez-Alonzo et al., 2013) and size-asymmetric competition both vary along secondary succession (Van Breugel, Martínez-Ramos & Bongers, 2006; Chazdon, 2014) and differentially affect large and small plants. Therefore, interacting effects of plant size, seasonality and successional age on SDTF dynamics can reasonably be expected. However, to our knowledge, our study is the first to assess plant size-dependent seasonal and successional dynamics (temporal changes in density, species density, recruitment and mortality) in a SDTF.

We use a chronosequence resampling approach over six years to examine successional and seasonal dynamics of demographic processes (recruitment and mortality) and of plant and species density of woody vegetation in two size classes: large (diameter at breast height, or dbh > 5 cm) and small (dbh 1–5 cm) plants in a SDTF in the Yucatan Peninsula, Mexico. We address the following questions: (1) How does community dynamics of woody plants vary between seasons and successional age categories? (2) (How) do large- and small-size plants differ in their successional and seasonal dynamics? We hypothesize that the successional patterns of plant and species density, and the underlying demographic rates: (H1) differ between seasons, reflecting successional and seasonal differences in key biotic (competition) and abiotic (water availability) factors; (H2) differ between large and small plants, reflecting size-dependent capacities to acquire, store and compete for limiting resources, and to withstand biotic and abiotic stressors. We predict that (P1) for large plants, plant and species density will rapidly increase early in succession (reflecting early colonization), especially during the wet season (when water availability is less limiting), and gradually stabilize in intermediate and advanced stages (due to increased competition in dense-cover, closed-canopy conditions); and that (P2) for small plants, after an initial increase, plant density will decline at intermediate and advanced successional stages (reflecting size-asymmetric competition), especially during the dry season (when water availability is most limiting), whereas species density will gradually stabilize (reflecting lower dominance and higher diversity). In line with these expected patterns of density, we predict that (P3) in early succession, recruitment and species gain rates will be high and mortality will be low for both large and small plants (especially in the wet season), while (P4) at intermediate and late successional stages all rates will be low for both large and small plants (as plant and species density gradually stabilize) except for (P5) high mortality rates of small plants (especially in the dry season).

Materials & Methods

Study area

The study was conducted within and around Kaxil Kiuic Biocultural Reserve (KKBR) in the center of the Yucatan Peninsula, Mexico (Fig. 1). Permission to access and conduct research in KKBR was granted by its Director, James M. Callaghan. Access to field sites on communal lands outside KKBR was verbally approved by don Evelio Uc Uc on behalf of the community of Xkobenhaltun. The climate is tropical, warm, subhumid, of the Aw type according to the Köppen-Geiger climate classification (Peel, Finlayson & McMahon, 2007). Mean annual precipitation is 841 mm, but annual precipitation varies greatly from 703 mm to 1,092 mm, with over two thirds of the yearly total falling during the rainy season (June–October), followed by a dry season with scattered rains (16–76 mm per month) from November to May (Fig. S1; Orellana et al., 1999). Mean annual temperature is 25.1 °C, but seasonal and monthly variation is high, with mean maximum temperatures of 36.1 °C at the end of the dry season (Jackson et al., 2018), which results in high vapor pressure deficit (mean of 2 kPa, compared to the wet season (mean of 0.5 kPa) (Cach-Pérez, Andrade & Reyes-García, 2018). The predominant vegetation is seasonally dry semi-deciduous tropical forest in which 50–75% of plants shed their leaves during the dry season. Canopy height ranges between 13–18 m, although vegetation structure varies with stand age, topography and soil properties (Dupuy et al., 2012). The study area has been subjected to extensive logging and clearing for traditional slash and burn agriculture, which requires a constant rotation of croplands due to the loss of soil fertility after two or three crops (Lara Ponce, Caso Barrera & Aliphat Fernández, 2012; Gunn, 2015). Consequently, the landscape is a diverse and shifting mosaic of croplands and forest stands of different fallow ages (Rico-Gray & García-Franco, 1991).

Figure 1 Location of the study area showing the spatial arrangement of sampling sites within and near Kaxil Kiuic Biocultural Reserve in the Yucatan Peninsula, Mexico, for each successional age category.

Sampling design

Nine permanent plots (20 × 50 m) were established in September and October 2009 in sites of similar conditions, which differ in fallow age. All plots are located on forest stands on flat areas with similar soil type (clayey Luvisols and Cambisols), subjected to traditional slash and burn agricultural practices within a fairly continuous matrix of secondary forests. Sites were used for fuel-wood extraction (mostly dead and/or fallen branches and stems) and experienced occasional poaching during forest regrowth.

Stand age (i.e., the number of years since a stand was abandoned) was determined from interviews with local residents who had lived in the area for at least 50 years and with knowledge of how the land was used by their parents in the decades prior to their birth. Mayan peasants in the Yucatan have developed their own detailed classification of successional stages, which is tightly linked to their traditional “milpa” system of slash-and-burn agriculture (González-Cruz et al., 2015), enabling a fairly accurate assessment of successional age of secondary forests. We categorized our study sites into three successional-age categories: (1) 3–5 year-old stands (hereafter named early-stage); (2) 18–20 year-old stands (intermediate stage); (3) >50 year-old stands (advanced stage; Table S1).

Within each 20 × 50 m plot, we recorded, tagged, identified and measured woody plants >5 cm dbh (mostly canopy trees, hereafter large plants) in ten 10 × 10 m quadrats, and plants 1–5 cm dbh (small—mostly understory—plants) in nested 5 × 6 m quadrats. Specimens of plants that could not be identified in the field were collected and identified using a regional reference collection from the CICY herbarium at the Centro de Investigación Científica de Yucatán.

Plants were initially measured at the end of the rainy season (September-October) of 2009 and approximately every six months (April-May, that is, at the end of the dry season) thereafter until 2015, to capture seasonal and inter-annual variation in woody plant community dynamics. Thus, following the initial inventory, a total of 12 censuses were conducted: six in the rainy season and six in the dry season. During these censuses, we measured the diameter of each stem and recorded individuals that had died or recruited into each plant size-category.

Statistical analysis

We assessed plant density (number of individuals/ha) and species density (number of species/plot), using the total area of 0.1 ha for large-size plants and 0.03 ha for small-size plants. For each plot, we calculated recruitment, mortality, species gain, and species loss rates in each census along the 6-year study period (2010 to 2015). These calculations considered plant size categories separately and included the values of the first census (the wet season of 2009) as the initial community. Seasonal demographic rates were calculated every six months (182.625 days), taking into account the periodicity of the censuses.

Plant recruitment rate was calculated as: (1) R=n+r∕n182.625∕t−1

where n is the number of plants present at the beginning of the season, r is the number of new recruits into the size class recorded in the corresponding census and t is the number of days elapsed between two consecutive censuses.

Plant mortality rate was calculated as: (2) M=1−1−d∕n182.625∕t

where n is the number of plants present at the beginning of the season, d is the number of plants that died between the previous census and the current one. Gain and loss rates of species were calculated using similar formulae, where n is the number of species, r the number of new species, and m the number of species lost from the previous census to the current one (Capers et al., 2005; Maza-Villalobos, Poorter & Martínez-Ramos, 2013). We used linear mixed-effects models to assess the effects of successional stage, seasonality and their changes over time on each response variable. We included time (year) as a predictor variable, since we expected each response variable as well as the effect of seasonality (i.e., variation between the wet and the dry season) to vary over time, even within each successional stage.

Because we had data from three plots of the same successional stage with repeated measures over six years, we included plot as a random effect. We modeled a random intercept and a slope for successional stage meaning the year-one dry-season response variables could vary among the three plots nested within each successional stage: advanced (intercept), intermediate (level 1) and early (level 2). Since the inclusion of plant size class in a single model along with the other predictor variables would result in an unwieldy model with a four-way interaction and all the sub-interactions, and considering that our dataset is too small to support a model with so many parameter estimates, we used separate models for the two plant size categories: small plants (1–5 cm dbh) and large plants (>5 cm dbh). Thus, we ran 12 models: six response variables (plant density, species density, recruitment, species gain rate, mortality and species loss rate) for the two size classes.

We related each community variable (Communityij) to the fixed explanatory variables using the following model: (4) Communityij=α+β1Y eari,j+β2Seasoni,j+β3Successional stagei,j+β4Yeari,jSeasoni,j+β5Y eari,jSuccessional stagei,j+β6Seasoni,jSuccessional stagei,j+β7Y eari,jSeasoni,jSuccessional stagei,j.

Where i is the plot, j is the census year, α is the intercept and β is the parameter estimate. Thus, we included 12 parameter estimates, corresponding to the 12 levels of the interactions among two seasons, three successional stages and year (advanced stage × dry season [the overall intercept, α], advanced × dry × year, advanced × wet season, advanced × wet × year, intermediate stage × dry, intermediate × dry × year, intermediate × wet, intermediate × wet × year, early stage × dry, early × dry × year, early × wet, early × wet × year). We estimated the parameters of each model (α, β1…) with the lmer function in the lme4 package (Bates et al., 2015). Additionally, we used the lmerTest package (Kuznetsova, Brockhoff & Christensen, 2017) to add denominator degrees of freedom and p-values to the table of coefficients based on the standard settings of lmerTest (Satterthwaite’s approximation). We present the results graphically by calculating the predicted seasonal trends in our variables along a six-year period in each of the three successional stages. Thus, by having the continuous variable, year, interact with the categorical variables, successional stage and season, we allowed the slope of the time series to vary by successional stage and season. We re-scaled the x-axis to show the stand age by adding each consecutive census year to the age of each plot at the initial inventory (rainy season of 2009), which we set as 5 years for the early stage, 20 years for intermediate and 60 years for advanced, although we acknowledge that these ages are averages and the real age of each stand may be 1–5 years older or younger. Fitted values and 95% confidence intervals for predicted values of models were obtained using parametric bootstrapping (n = 999) within the bootMer function in the lme4 package (Bates et al., 2015) and visualized within the R package ggplot2 (Wickham, 2016).

For each response variable, we calculated the marginal (m) and the conditional (c) R2 values using the r.squaredGLMM function in the MuMIn package (Bartoń, 2018). Rm2 indicates the variance explained by fixed effects only and R2 c indicates the variance explained by both fixed and random effects (Nakagawa & Schielzeth, 2013). All analyses were performed in R version 3.6.0 (R Development Core Team, 2012).

Results

Plant and species density

A total of 4,319 individual plants belonging to 120 species and 34 families were recorded in the total sampling area (0.9 ha) during the six-year study period. Large plants (dbh > 5 cm) represented 32% of individuals in the community and encompassed 53 woody species in 18 families. Small-size plants (dbh 1–5 cm) accounted for 68% of the recorded individuals and belonged to 113 species distributed in 33 families.

Mean (±SE) plant density increased with successional stage from 682.5 ± 316.2 plants/ha in the early stage, to 1,320.5 ± 235.7 in the intermediate stage and 1,538.3 ± 272.7 plants/ha in the advanced stage. The mean density of large plants increased markedly in the early stage from 288 ± 81 plants/ha in the first study year (2010), to 1,132 ± 174 plants/ha in the last year and was higher in the wet than in the dry season, but seasonal differences were vastly reduced and plant densities stabilized in intermediate and advanced stages (Table S2; Figs. 2A–2C). Species density of large-size plants showed a similar pattern, increasing with stand stage, and having a pronounced increment during the study period only in the early stage, but did not vary between seasons at any stage (Table S2; Figs. 2D–2F).

Figure 2 Predicted temporal trends for large plant density (A–C) and species density (D–F), and for small plant density (G–I) and species density (J–L) in the dry- and rainy seasons in each successional stage.

The shaded sections correspond to 95% confidence intervals. Asterisks in each graph indicate significant (p > 0.05) effects of predictors based on the linear mixed effects models. D*, dry season; W*, wet or rainy season; Dry × Year*, dry season × sampling year interaction.

In contrast to large plants, density of small plants decreased over time, especially in the advanced stage, and also decreased with successional stage from 11992.5 ± 3087.0 plants/ha in the early stage, to 7821.2 ± 813.6 in the intermediate stage and 6335.1 ± 595.4 plants/ha in the advanced stage, but did not vary between seasons (Table S3; Figs. 2G–2I). Species density of small plants showed a very different pattern; it increased with successional stage and over time in the early stage but decreased over time in the advance stage (Table S3; Figs. 2J–2L).

Recruitment and species gain rates

Both recruitment and species gain rates of large plants were highest and showed interannual and seasonal variation exclusively in the early stage. At this stage, both rates were higher in the rainy than in the dry season at the beginning of the study but decreased thereafter, converging to dry season values. At the intermediate and advanced stages, recruitment and species gain rates were small and did not vary between seasons or among years (Table S4; Figs. 3A–3F). Recruitment and species gain rates of small plants were both very low and did not vary over the study period, between seasons or among successional stages (Table S5; Figs. 3G–3L).

Figure 3 Predicted temporal trends for large plant recruitment (A–C), species gain rate (D–F), small plant recruitment (G–I), species gain rate (J–L) in the dry- and rainy seasons per successional stage.

The shaded sections correspond to 95% confidence intervals. Asterisks in each graph indicate significant (p > 0.05) effects of predictors based on the linear mixed effects models. W*, wet or rainy season; Wet × Year*, wet season × sampling year interaction.

Mortality and species loss rates

Mortality of large plants was highest and showed significant interannual and seasonal variation only at the intermediate stage. At this stage, mortality was higher in the rainy than in the dry season at the beginning of the study and decreased thereafter converging to dry season values (Table S6; Figs. 4A–4C). Species loss rates of large-size plants were generally low and showed no significant effect of any of the predictors or interactions evaluated, partly due to large among-plot variation (Table S6; Figs. 4D–4F).

Figure 4 Predicted temporal trends for large plant mortality (A–C) and species loss rate (D–F), small plant mortality (G–I) and species loss rate (J–L) in the dry- and rainy seasons per successional stage.

The shaded sections correspond to 95% confidence intervals. Asterisks in each graph indicate significant (p > 0.05) effects of predictors based on the linear mixed effects models. D*, dry season; W*, wet season; Wet × Year*, wet season × sampling year interaction.

On the other hand, for small plants, mortality rate increased slightly with successional stage, with the highest values in the advanced stage (Table S7; Figs. 4G–4I). Finally, species loss rates of small-size plants showed large among-plot (random) variation and no significant effect of any of the predictors or interactions (Table S7; Figs. 4J–4L).

Discussion

Surprisingly, most studies of plant community dynamics along seasonally dry tropical forest (SDTF) succession are based on annual surveys, and hence fail to consider the potential effects of seasonality (Maza-Villalobos, Balvanera & Martínez-Ramos, 2011; Derroire et al., 2016; Rozendaal et al., 2017; Martínez-Ramos et al., 2018; but see Maza-Villalobos, Poorter & Martínez-Ramos, 2013). To our knowledge, this is the first study to assess seasonal and successional dynamics of size-dependent plant demographic rates in a SDTF. We asked how community dynamics of woody plants vary among seasons and successional age categories and if large- and small-size plants differ in their successional and seasonal dynamics. We hypothesized that the successional patterns of plant and species density, and the underlying demographic rates would differ both between seasons and between large and small plants. We found marked differences in demographic rates and in plant and species density between large- and small-size plants between seasons, among successional stages and over the study period, but significant differences between seasons only for large plants. Large plants exhibited high wet-season recruitment and species gain rates at the early stage and high wet-season mortality at the intermediate stage (both decreasing with time), resulting in an increase in plant and species density early in succession and a subsequent stabilization. In contrast, demographic rates of small-size plants showed large among-plot (random) variation and almost no significant effects of time (year), seasonality or successional stage. However, small plant density decreased steadily after only 5 years of land abandonment, whereas species density increased with successional stage, likely reflecting strong dominance early in succession and greater diversity at later stages. Below we discuss the results in detail.

We predicted that plant and species density of large plants would increase early in succession and stabilize subsequently (P1), reflecting high recruitment and species gain and low mortality and species loss rates (P3). Our results clearly matched these predictions (Figs. 2A–2F–4A–4F) and indicate that large-plant density and species density recover to old-growth forest values quickly, in line with previous findings in other SDTF (Aide et al., 2000; Lebrija-Trejos et al., 2008; Chazdon et al., 2011; Maza-Villalobos, Balvanera & Martínez-Ramos, 2011). Moreover, early successional recruitment and species gain rates were higher in the rainy than the dry season (Figs. 3A and 3D), which is in line with findings of previous studies in SDTF indicating that seasonal water availability enhances plant growth and establishment (Holbrook, Whitbeck & Mooney, 1995; Dirzo et al., 2011; Gaviria & Engelbrecht, 2015; Alberton et al., 2019; Campos et al., 2019). In SDTF, temperature, solar radiation and water vapor-pressure deficit are higher while soil water content is lower at the early successional stage compared to subsequent stages (Lebrija-Trejos et al., 2011; Pineda-García, Paz & Meinzer, 2013). Thus, early in succession, growth of saplings into the large size class is likely low during the dry season, occurring mostly during the wet season. Previous studies in SDTF have shown that many early successional species have a fast or acquisitive strategy maximizing water transport, photosynthesis and biomass accumulation when water is available, while minimizing water loss and respiration costs—hence, also photosynthesis and growth—by closing their stomata or shedding their leaves during dry periods (Méndez-Toribio et al., 2020; Subedi et al., 2019). In later stages, canopy closure and biomass accumulation may somewhat buffer the adverse effects of seasonal drought on plant growth, allowing recruitment and species gains of large plants—especially those with slow or conservative strategies—even during the dry season. In agreement with our findings, Bretfeld, Ewers & Hall (2018) found a stronger response to an ENSO dry-season drought—with reductions in water-use to avoid hydraulic failure—in early-successional compared to late-successional SDTF stands in Panama.

The combination of high recruitment and species gain rates with low mortality and species loss rates early in succession resulted in a fast increase in large-size plant and species density. Sixty six percent of large-size plants (287 out of 425) that recruited during the study period were recorded in early successional plots. In contrast, only 3% (4 out of 133) of the large-size plants that died during the study period did so during early succession. Recruitment in early successional plots was dominated by pioneer species, such as Leucaena leucocephala (Fabaceae), Mimosa bahamensis (Fabaceae), Heliocarpus donnellsmithii (Malvaceae) and Cochlospermum vitifolium (Bixaceae), and by generalist species, such as Piscidia piscipula (Fabaceae) and Bursera simaruba (Burseraceae). These six species combined accounted for 74% of all recruits in early-successional plots.

Our prediction (P4) that large plants would show low values of all demographic rates (recruitment, species gain, mortality and species loss) at intermediate and advanced successional stages was partly supported. Species gain and loss rates of large plants were indeed low at the intermediate and advanced successional stages (Figs. 3E and 3F; 4E and 4F), resulting in fairly stable species densities (Figs. 2E and 2F). However, compared to the other successional stages, mortality of large plants was higher at the intermediate stage, especially during the wet season (Fig. 4B), accounting for 62% of all large plants that died in this study. This roughly coincides with the timing of peak mortality found by Lebrija-Trejos et al. (2010a) in a SDTF in Oaxaca, Mexico: 15 to 18 years after abandonment, compared to 18 to 20 years in this study. These authors attributed this peak mortality (and a concomitant decline in the dominance of pioneer species) to competition within this guild—rather than competition with the mature forest guild. Canopy closure and the structural recovery of vegetation along succession are expected to increase the influence of competition and density-dependent processes on plant dynamics (Callaway, 1997; Bhaskar, Dawson & Balvanera, 2014; Sanaphre-Villanueva et al., 2017; Dalmaso et al., 2020). The higher mortality of large plants at the intermediate successional stage may thus be associated with competition and density-dependent processes (Niinemets, 2010; Sanaphre-Villanueva et al., 2016).

In our study site, three early-successional species, Neomillspaughia emarginata (Polygonaceae), Mimosa bahamensis and Bauhinia ungulata (Fabaceae), together accounted for 63% of all deaths recorded in intermediate successional plots. These combined results suggest that our intermediate successional stage may correspond to the stand exclusion stage, characterized by self-thinning of the dominant pioneer species (Chazdon, 2008; Chazdon, 2014). However, in our site, these dominant early-successional species did not show a sharp decline at the intermediate and advanced stages, due to their continued recruitment (i.e., growth of previously established seedlings and saplings), which partially compensated for their high mortality. In particular, sprouting may be an important mechanism allowing these species to persist through succession (Dupuy et al., 2012).

The higher mortality of large plants observed in the wet compared to the dry season at the intermediate successional stage was unexpected, since drought during the dry season has been associated with reduced growth and increased plant mortality (Swaine, Lieberman & Hall, 1990; Lieberman & Li, 1992; Gerhardt, 1993; Ceccon, Huante & Rincón, 2006; Campos et al., 2019; Marques et al., 2020). Leaf fall reduces the ability to unequivocally identify plant mortality during the dry period. In other words, during the dry season, it is difficult to determine whether a plant is actually dead, since drought-deciduous plants may appear dead but flush new leaves in the following wet season. To control for this possibility, we conducted a separate analysis of large-plant mortality only for evergreen species—for which dry-season mortality can be more confidently determined. We found similar patterns and the same qualitative results shown by all large-size plants (Figs. 4A–4C, Figs. S2A–S2C), suggesting that our results may not be attributed to a sampling bias.

In a SDTF in Jalisco, Mexico, Maza-Villalobos, Poorter & Martínez-Ramos (2013) found higher mortality of seedlings in the wet than in the dry season. Noy-Meir (1974) first proposed the pulse-reserve paradigm for arid and semi-arid environments, where resources such as water and nutrients are available during pulses and unavailable during the interpulses, when plants adapted to these environments survive on carbon and water reserves. Goldberg & Novoplansky (1997) proposed that, in productive environments, competition would be the most important factor limiting species persistence, while in environments with low productivity and prolonged interpulses, survival during interpulses would drive species persistence. Our results suggest that the community dynamics in the SDTF of our study are driven mostly by competition during pulses (wet seasons). Nevertheless, the interpulse (dry season) may still play a role in plant mortality; first, plants may reduce or deplete non-structural carbohydrates during the dry season to withstand drought (Amthor & McCree, 1990; Myers & Kitajima, 2007), which may lead to increased mortality in subsequent rainy seasons. Moreover, with the onset of rain, plants have to invest significant carbon reserves to produce new leaves (deciduous species) or repair damaged structures such as the photosynthetic apparatus (evergreen species) further depleting non-structural carbohydrates (Schwinning & Weiner, 1998; Reyes-García & Griffiths, 2009). Second, reduced or depleted reserves may render plants more vulnerable to biotic stress from herbivores and pathogens or from shading and root competition—all of which peak during the rainy season (Filip et al., 1995; Cuevas-Reyes, Quesada & Oyama, 2006; Anderegg & Callaway, 2012). This may jeopardize the capacity to keep a positive carbon balance to maintain plant respiration and growth, resulting in death. The duration and intensity of pulses and interpulses in water and nutrient availability will likely be affected by climate change through shifts in rainfall patterns, such as increases in the duration and severity of dry-season drought (interpulse), or reductions in the amount of rainy season precipitation (pulse)—(Allen et al., 2017; Stan & Sanchez-Azofeifa, 2019). Although our study is admittedly too short to draw any conclusions, our results suggest that successional dynamics in our study forest may be as (or even more) strongly influenced by climate-change driven decreases in wet season rainfall as (than) prolonged or severe droughts during the dry season.

Our prediction (P2) that small plant density would increase early in succession and subsequently decline, whereas species density would stabilize following an initial increase was partly supported by our results. Small plant density declined steadily not only at the intermediate- and advanced stages, but also at the early successional one, after only 5 years of land abandonment (Figs. 2G–2I). This steady decline in small plant density after such short abandonment period, coupled with the opposite successional patterns of density of small (decreasing) vs. large (increasing) plants, suggest a potential role of competition amongst small plants—early in succession—as well as with larger plants—at intermediate- and advanced successional stages (Niinemets, 2010; Sanaphre-Villanueva et al., 2016). A previous study in the same study region also found opposite successional patterns of density for small vs. large plants and suggested size-asymmetric competition as a potential cause (Dupuy et al., 2012).

We found only partial support for our prediction (P5) that, at intermediate and advanced successional stages, mortality of small plants would be high and recruitment low, leading to a decline in density, whereas species gain and loss rates would be generally low, thereby stabilizing species density after an early-successional increase. Recruitment of small plants was low and did not vary over succession (Figs. 3G–3I), while mortality was slightly higher at the advanced successional stage (Figs. 4G–4I). Dalmaso et al. (2020) also found a net negative balance between recruitment and mortality leading to a reduction in plant density in advanced successional stages of a Brazilian Atlantic forest, likely reflecting intense competition. In our study, small plant density declined steadily after only 5 years of land abandonment (Figs. 2G–2I), implying a fairly constant negative balance between recruitment and mortality in all successional stages, as also found in other Mexican SDTF (Maza-Villalobos, Balvanera & Martínez-Ramos, 2011). Since plant size increases over succession and small plants dominate density patterns (Dupuy et al., 2012), our results indicate that overall plant density peaks very early in succession, creating a short, dense, fairly uniform canopy. This may attenuate the effect of drought, which could help explain the generally low dry season mortality rates observed in this study. A recent long-term study in European temperate forest showed that increasing canopy cover can reduce warming rates inside forests and can decouple local interior (microclimatic) conditions from regional (macroclimatic) ones outside forests, thereby buffering against global warming (Zellweger et al., 2020). On the other hand, canopy closure may also increase the influence of competition and density-dependent processes on plant dynamics (Callaway, 1997; Bhaskar, Dawson & Balvanera, 2014; Sanaphre-Villanueva et al., 2017). Thus, the slightly higher mortality of small plants at the advanced successional stage may be associated with competition and density-dependent processes (Niinemets, 2010; Dalmaso et al., 2020).

The absence of significant differences in species gain or loss rates of small plants among successional stages coupled with the increase in species density with successional stage imply a positive balance between species gain and loss rates. The increase in species density of small plants is in line with previous findings that species density increases along SDTF succession (Guariguata & Ostertag, 2001; Ruiz, Fandiño & Chazdon, 2005; Lebrija-Trejos et al., 2008; Dupuy et al., 2012). This increase in species density, however, contrasts with the decrease in small plant density across succession. These contrasting patterns are consistent with a strong dominance of the small plant community by a few species with very high abundance early in succession, and a subsequent decline in dominance (and density), allowing other species to colonize and increase species density. Only three pioneer species (N. emarginata, M. bahamensis and Helicteres baruensis—Malvaceae—) accounted for almost 60% of total abundance in the early successional stage. In the late successional stage, the small plant community showed lower dominance by a mix of generalist species (Eugenia axillaris—Myrtaceae—, Guettarda gaumeri—Rubiaceae—) and late-successional ones (Psidium sartorianum—Myrtaceae—, Amphilophium paniculatum and Bignonia neoheterophylla—Bignoniaceae—), which together accounted for 44% of total abundance. Dupuy et al. (2012) also documented a general decline in woody plant density and species dominance coupled with an overall increase in plant species richness across forest succession in the same study region.

The effects of seasonality, successional stage and their change over time were more apparent on large-size plants than on small-size ones (Figs. 2–4). This result was somewhat unexpected, since small plants have smaller root and shoot systems to capture resources and store carbon and water, and therefore were expected to be more vulnerable to seasonal drought and more dependent on seasonal water availability for recruitment and growth than large plants (Kitajima & Fenner, 2000; Quitete-Portela & Maës dos Santos, 2009). It is likely that the large among-plot variation shown by small plants, coupled with strong annual variation in rainfall (Fig. S1) obscured the effects of seasonality, successional stage and time. It is also possible that the strongest successional and seasonal-drought filters occur in plants smaller (i.e., dbh < 1 cm) than those included in this study. In our study forest, Jackson et al. (2018) described strategies for the survival of small trees, such as deep roots, osmotic adjustment and tight stomatal control. Alternatively, recruitment, mortality and species gain and loss rates of small plants could be more strongly influenced by other factors that may confound the effects of seasonality and successional stage. For example, compared to larger plants, dispersal limitation can be expected to have a greater influence on small plants (in earlier stages of plant ontogeny) thereby potentially increasing random among-plot variation. The consistently lower values of marginal variance (R2m)—explained by fixed effects only—and the generally greater relative difference between conditional (R2c) and marginal variance—indicating the importance of random effects—found for small plants compared to large ones (Tables S2–S7) are consistent with this interpretation. Alternatively, local site factors, such as con-specific density, the spatial distribution of neighboring plants or the identity of the dominant species may alter biotic interactions (e.g., competition, herbivory) and/or abiotic conditions (e.g., light and water availability) and thereby affect the demographic rates of small plants (Granda, Escudero & Valladares, 2014; Mesquita et al., 2015; Espinosa et al., 2016; Ma et al., 2016), which may be particularly sensitive to such local factors. Recent studies show that interactions among multiple sources of environmental stress (climatic and soil variables) play a key role in plant species filtering—which may be expected to have a greater impact on small than on large plants—in dry forests (Bagousse-Pinguet et al., 2017; Méndez-Toribio et al., 2020). Further studies are needed to elucidate the complexity of ecological factors and plant strategies that influence plant and species density as well as seasonal and successional size-dependent dynamics and demographic rates in SDTF.

Conclusions

As hypothesized, we found marked differences in demographic rates and in plant and species density between large- and small-size plants across succession, between seasons and over the study period. Clear trends in how these attributes changed across years, seasons, and successional stages were more apparent for large than for small plants. Small plant density decreased steadily after only 5 years of land abandonment, whereas species density increased with successional stage, likely reflecting a decline in species dominance. Overall, we found no clear successional, interannual or seasonal trends in small plant demographic rates likely due to large random among-plot variation. In contrast, seasonality affected demographic rates and density of large plants, which exhibited high wet-season recruitment and species gain rates at the early stage coupled with high wet-season mortality at the intermediate stage, resulting in the predicted increase in plant and species density early in succession and subsequent stabilization. Notwithstanding the short duration of our study, our results suggest that a climate change-driven decrease in rainy season rainfall may have an influence on successional dynamics (and hence resilience) in our study forest as strong as, or even stronger than, prolonged or severe droughts during the dry season.

Supplemental Information

Table S1 Successional (forest stand) age and category of each study plot

Click here for additional data file.

Table S2 Estimates for mixed effects models of plant density and species density of large plants

Significant P values (≤0.05) are indicated in boldface. The standard errors (SE), conditional R2 (R2c, both fixed and random effects), and the marginal R2 (R2m, fixed effects only) as well as the relative (%) difference between them (indicating the importance of random effects) are shown.

Click here for additional data file.

Table S3 Estimates for mixed effects models of plant density and species density of small plants

Significant P values (≤0.05) are indicated in boldface. The standard errors (SE), conditional R2 (R2c, both fixed and random effects), and the marginal R2 (R2m, fixed effects only) as well as the relative (%) difference between them (indicating the importance of random effects) are shown.

Click here for additional data file.

Table S4 Estimates for mixed effects models of recruitment rate and species gain rate of large plants

Significant P values (≤0.05) are indicated in boldface. The standard errors (SE), conditional R2 (R2c, both fixed and random effects), and the marginal R2 (R2m, fixed effects only) as well as the relative (%) difference between them (indicating the importance of random effects) are shown.

Click here for additional data file.

Table S5 Estimates for mixed effects models of recruitment rate and species gain rate of small plants

Significant P values (≤0.05) are indicated in boldface. The standard errors (SE), conditional R2 (R2c, both fixed and random effects), and the marginal R2 (R2m, fixed effects only) as well as the relative (%) difference between them (indicating the importance of random effects) are shown.

Click here for additional data file.

Table S6 Estimates for mixed effects models of mortality rate and species loss of large plants

Significant P values (≤0.05) are indicated in boldface. The standard errors (SE), conditional R2 (R2c, both fixed and random effects), and the marginal R2 (R2m, fixed effects only) as well as the relative (%) difference between them (indicating the importance of random effects) are shown.

Click here for additional data file.

Table S7 Estimates for mixed effects models of mortality rate and species loss of small plants

Significant P values (≤0.05) are indicated in boldface. The standard errors (SE), conditional R2 (R2c, both fixed and random effects), and the marginal R2 (R2m, fixed effects only) as well as the relative (%) difference between them (indicating the importance of random effects) are shown.

Click here for additional data file.

Figure S1 Monthly and seasonal patterns of precipitation and temperature in our study area over the period 2006-2016

The graph shows mean ± SE monthly rainfall and temperature, while the table shows mean, minimum (Min), maximum (Max) and coefficient of variation (CV) values of each variable in the dry and wet seasons.

Click here for additional data file.

Figure S2 Predicted temporal trends over the study period for mortality of large plants of perennial species in the dry- (red circles) and rainy (blue triangles) seasons in each successional stage (early, intermediate and advanced)

The shaded sections correspond to 95% confidence intervals. Capital letters in each graph represent significant (p < 0.05) effects of predictors based on the linear mixed effects models. W*: wet season, Wet × Year*: wet season × sampling year interaction.

Click here for additional data file.

Figure S3 Diagnostic plots for plant and species density models of large (a–d) and small (e–h) plants

Left panels (a, c, e and g) show the distribution of standardized residuals with respect to normal quantiles to assess the normality assumption. Right panels (b, d, f and h) show residuals against fitted values to assess the linearity assumption.

Click here for additional data file.

Figure S4 Diagnostic plots for recruitment and species gain models of large (a–d) and small (e–h) plants

Left panels (a, c, e and g) show the distribution of standardized residuals with respect to normal quantiles to assess the normality assumption. Right panels (b, d, f and h) show residuals against fitted values to assess the linearity assumption.

Click here for additional data file.

Figure S5 Diagnostic plots for mortality and species loss models of large (a–d) and small (e–h) plants

Left panels (a, c, e and g) show the distribution of standardized residuals with respect to normal quantiles to assess the normality assumption. Right panels (b, d, f and h) show residuals against fitted values to assess the linearity assumption.

Click here for additional data file.

Data S1 Response variables of large- and small-size plants in each successional age category

Plant density, species density, recruitment, mortality, species loss rate and species gain rate of each sampling plot (arranged by successional age category) in each consecutive census, indicating the season (dry or wet), as used for all statistical analyses.

Click here for additional data file.

We thank James Callaghan and Kaxil Kiuic A.C. for the facilities and support provided, and the local inhabitants of Xkobenhaltun for their continued help with fieldwork.

Additional Information and Declarations

Competing Interests

Author Contributions

Field Study Permissions

Data Availability

The authors declare there are no competing interests.

Irving Saenz-Pedroza performed the experiments, analyzed the data, prepared figures and/or tables, authored or reviewed drafts of the paper, and approved the final draft.

Richard Feldman analyzed the data, prepared figures and/or tables, authored or reviewed drafts of the paper, and approved the final draft.

Casandra Reyes-García and Filogonio May-Pat performed the experiments, authored or reviewed drafts of the paper, and approved the final draft.

Jorge A. Meave analyzed the data, authored or reviewed drafts of the paper, and approved the final draft.

Luz Maria Calvo-Irabien conceived and designed the experiments, authored or reviewed drafts of the paper, and approved the final draft.

Juan M. Dupuy conceived and designed the experiments, performed the experiments, prepared figures and/or tables, authored or reviewed drafts of the paper, and approved the final draft.

The following information was supplied relating to field study approvals (i.e., approving body and any reference numbers):

Permission to access and conduct research in Kaxil Kiuic Biocultural Reserve (KKBR) was granted by the Director of KKBR, James M. Callaghan. Access to field sites on communal lands outside the KKBR was verbally approved by don Evelio Uc Uc on behalf of the community of Xkobenhaltun.

The following information was supplied regarding data availability:

The raw data are available in Data S1.

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
