# Peer review of "Seasonal and successional dynamics of size-dependent plant demographic rates in a tropical dry forest"

_PeerJ, doi:10.7717/peerj.9636_

## Round 0.1 · original submission · Major Revisions

The three reviewers considered your paper very interesting and worth publishing. However a number of issues needs to be resolved, among the most important are to propose a stronger hypothesis based on your conceptual framework. In addition it will be important to provide more information on climate patterns of the area of study. Figures need improvement (see below) as well as explaining better figure legends. With regard to methods additional details on the predictions, models and stages are needed. Below you will find all these issues.

Reviewer 1 ·

Basic reporting

The article is clearly written, the literature is good and vast, although it would be good to incorporate a little more recent literature. Figure 1 has very poor quality and is not clearly located in the geographical context. The graphs should explain or know what the gray lines are. Figures 2, 3 and 4, try to improve them, or put the results in tables. The hypotheses are quite evident.

Experimental design

No comments

Validity of the findings

No comments

Additional comments

The hypotheses and predictions are very obvious, an effort should be made to understand the ecological mechanisms behind the approach. It is not clear why they divide between small and large trees, and the inferences they make about the effect of climate change are very risky because they only have six years of data.

·

Basic reporting

1. General writing can be improved. There are sections with very large paragraphs, making it difficult for the reader to focus and identify the main ideas. See particularly lines 72-118 in the introduction section and lines 380-451 of the discussion. For the introduction section, I suggest authors to split those lines into two to three paragraphs, focused on 1) current knowledge of successional patterns of density, species, density, and demographic rates, 2) what is known about differences in seasons and why it is expected to modify those patterns, and 3) the effect of plant size on those patterns and the possibility of interaction with season. For the discussion section, it seems that by simply splitting the current paragraph at lines 390 and 408 would be enough. However, please begin each of these paragraphs with a line describing the main idea to be discussed.

Also, in lines 125-127 hypothesis should be rephrased. Authors say: “We hypothesize that the successional patterns of plant and species density, and the underlying demographic rates will differ between seasons as well as between large and small plants”. For it to be sound ecological hypothesis it should propose some mechanism(s) from which such a result is expected, rather than simply expressing an expectation. Authors have provided background on previous lines of the introduction, so they just have to summarize it here.

2. Authors have made an exhaustive bibliographic review of the existing literature for tropical forests on the topics addressed in the manuscript. However, in order to bring their results to a broader context, such as that of tropical forests in general, authors could search for references from other kind of systems and discuss their results in such a context. Otherwise, it is not clear how generalizable or not are their results.

In line 95 Mendez-Alonso et al. 2013 may not be the adequate citation here, as they do not assess successional gradients. Probably better to cite Pineda et al. 2013 Plant, Cell & Environment. DOI: 10.1111/j.1365-3040.2012.02582.x. Also in line 282. Maza-Villalobos et al., 2013 did actually assessed the effects of seasonality in their work. Please be specific or highlight here (and in the introduction section) that this is the only work doing that (although they did not assess the effect of different size classes).

3. Figures can be modified to improve clarity in the presentation of results. Main suggestions are:
- Instead of presenting two different variables in a graph but for the same size class, I suggest authors present all the results for the same variable in a single graph. That would allow the reader to directly compare the results for different size classes and not moving between plots for making such comparisons.
- Although presenting the results as the prediction from the fitted models is a good idea, lacking the data in the figures do not allow readers to graphically assess how well the model fitted to the data. So please present the original data as points in the figures.
In all figures, I do not understand why authors present significance of both seasons. - Season is categorical predictor with two levels. So in the model one of the seasons should be included as part of the “basal” level (defining the intercept) and the other defines a contrast in relation to that level. So it should be there only one test and P-value associated with the season main effect. When looking at Fig_S1 it seems that what the authors are reporting as the effect of “Dry_season” is a test of the null that the intercept (i.e. the dry condition) is 0. That is not a relevant test in this case, and as such only the test for the wet period (which is actually a test for the difference between dry and wet main effects) should be reported.
- Supplementary Figures must be checked, as they seem to have some errors. For example, Figure_S1 reports a negative effect of dry season at early stage in both panels a and b. However, it seems those effects are positive, as deduced from the values reported above the error bars but also from the general coherence of the results. Similar problems are presented in other supplementary figures. Please double-check all the model parameter estimations and the values reported in these figures.

4. The article is a self-contained work, with all the relevant results to test the proposed hypotheses.

Experimental design

1. The article is without doubt primary research.

2. Research questions are clearly stated at the last paragraph of the introduction. I consider they are interesting and relevant for those working on forest recovery in tropical landscapes. However, due to the lack of clarity while framing the current knowledge, knowledge gaps and relevance of the work in the previous paragraphs of the introduction, I consider it will not be clear for the general audience why the work is relevant and meaningful. As suggested before, writing of the introduction needs to be improved. See my previous comments on how to do it.

3. Sampling methods are detailed enough. However, in line 141 please specify if sites were excluded from human use during forest regrowth or if they are used for some kind of forest products extraction or cattle ranching.

Regarding analytic methods there are some details to be further explained and other may need to be modified:

- Given that one of the main purposes of the work is to assess seasonal variation in community properties and demographic rates and how they change with successional stage/age and plant sizes, it would be ideal to fit a single model including all these predictors. However, authors fit independent models for different plant size classes. This approach does not allow to formally test the interaction between size class and season or successional stage, which is a central point of the second question and included in all the predictions presented. Therefore, I suggest the authors fit a complete model. If it is not possible because not enough data is available, please say so in the text.
- I find questionable to nest the random effect of site within the successional category. The random effect of site is included to account for the non-independency and therefore a possible systematic effect of site on response variables. Such an effect should in principle not be related with successional category and if so, it would be accounted by the fixed effect of successional category. So It does not seem to be necessary and in fact could make the model harder to fit. So please consider to remove successional category as a nesting variable in the definition of such random effect.
- Also regarding random effects, please consider to include the sampling date as second random effect orthogonal to the site effect. This is because observations from the same date in different sites are not truly independent: they could all be biased on the same direction if, for example, the same uncalibrated measurement tool (or observer) was used across all the plots in that sampling date. It may not be relevant, but you do not know that in advance. lmer allows for crossed random effects.
- It is not clear if random effects were applied only on the intercept of the model or also on the slope of time or some other fixed predictor. It seems it was only on the intercept, but please specify. Authors can add random effects in the model formula (formula 3) for clarity.
- Please provide a description of which kind of hypothesis tests are presented: are they single parameter tests or tests of complete effects (Anova-type tests)? As lmer function does not provide such tests, please specify which library and/or function did you use.
- In lines 205-206 authors explain they include time as a fixed predictor to take into account the possibility that the effect of seasonality may vary over time. Seeing the results, it seems that the main reason to include time as a fixed predictor is because response variables change over successional time, particularly in early successional sites, so not including time would make the model to fall short in representing data.

Validity of the findings

1. Results are fairly valid. However, claim made in lines 452-455 of the discussion and repeated in lines 469-472 of the conclusions section does not seem to be well linked to results or the discussion. This lines are disconnected from previous discussion and therefore not well supported. Please try to provide further support for them, for example, by returning to lines 69-71 in the introductory section.

2. One way to assess the validity of the findings in a work like this, where statistical modelling is central, is to provide at least a graphical summary of the adequacy of the models fitted. Diagnosing models through tools like a graphical summary of the residuals (their relation to predicted values or if their distribution is close to normal) is commonly enough to assess model validity. So please provide, as supplementary material such a diagnosis.

Reviewer 3 ·

Basic reporting

Review on “Seasonal and successional dynamics of size-dependent plant demographic rates in a tropical dry forest (#44799)” by Saénz et al.

The manuscript presented by Saénz et al. on secondary succession of TDF in the Yucatán Peninsula shows the importance to consider, not only the effects of successional stage, but also the seasonality, which is characteristic of this important forest in the neotropics. The authors also show elegantly that size of plants display different responses, which can be explained on limiting-resources competence basis.
Overall, I found the manuscript very interesting, however, there are some issues, which I´d like to see addressed in a new version. Below, the authors will find some observations/suggestions, which the author may consider the recommended new version.

Introduction
L75-77. Can you expand a little this part of the introduction? Or why the authors argument between wet and dry forest?
L123-124. Be consistent with how/How.
L125. The hypothesis sound very simplistic. Can you elaborate a stronger hypothesis based on your conceptual framework?
L127. Is “P1A” a part of the prediction 1? I suggest to number it or explain what does “P1A” stand for?

Experimental design

Methods
L142. The map does not need state division. I suggest to include the border limits with countries at North and South and other landmarks, at least the names of Ocean Pacific, Gulf of Mexico, United States and Guatemala/Belize. You definitively may not believe it, but some people in the other side of the world do not know the location of Mexico.

L143-149. Given the rainfall and temperature are very relevant for their study, I suggest authors to include more detailed information on climate patterns. For example, authors can provide total rainfall for the rainy and dry season and they can provide an estimation of interannual variation. I do not understand what the authors mean with “mean annual precipitation ranges” the mean annual precipitation should be only one number. If they provide two numbers this is a range, not a mean.
Authors say that most of the rainfall falls during the rainy season, but how much? The same should be presented for temp. May I recommended to explore the relationship of your response variables with the rainfall/temp of the dray and wet season, instead the seasonality (as a categorical variable).

Sampling design
Do the authors have information for rainfall for the wet and dry season for the six years of the study? Is there any way you can use this information for the analysis, instead just the categorical variable (seasonality)?

L160-167. Authors should include a table with information for this site, including the specific age obtained from interviews to local residents.
L165. It sound odd that local residents living for at least 40 yr. can give information for stands >50 years old stands (and from the plots in results, I can see there are ages of 65 yrs). Local residents have been living there for at least 65 ys., right? Probably more, but you have evidence that at least they´ve been living there for 65 ys.
L172. I suggest to name your institution in Spanish. Only my opinion.
L177. How was the interannual variation of temp and rainfall in the six years? Why do the authors include seven censuses in the rain and six in the dry season? They probably can include in an Appendix a plot showing the rainfall and temp variation during six years.
L203-206. Not sure to understand why year was included as a fixed effect? How successional stage and seasonality was included in the model? Are not they fixed effects too? Looking at the model, I believe they are fixed effect? If they are not, what are they then?

L215-224. This is indeed a complex model. Did the authors remove non-significant terms? Or were included in the model, even when they are not significant. The authors should explain how they did use their model to predict and plot the predicted values as a function of age.
Also in results and plots, authors refer to “Time”. Are they referring to Successional stages?
If Succesional stage was included in the model as categorical variable, I cannot understand how the models (and plots) were obtained, as “Age” is numerical.

Can the authors explain how they make the prediction for each successional stage as a function of age? Did they make a model for each stage? Why each successional stage have different ages (5-10, 20-25 and 60-65) than those of the included in each stage (3-5, 8-20 and >50 ys).

Validity of the findings

The findings are interesting, however, I believe the author may improve their finding with the suggestions made. Below there are some suggestions for the Results section,

Results
How do the authors make the predictions and produced their figures? Authors should explain more detailed.
“Succesional stage” is a categorical variable, which is not reflected in the plots or at least, authors do not explain.
It is not clear to me, why each panel (early, intermediate and advance) do have their own prediction (I mean why is not a continuum sequence)? Did they produce a model separately? Can the authors to explain all these details?

What are the shaded sections in all figures? Are they confidence intervals? Authors should explain.

The legend of all figures is confused, you may be want to include the R2m and R2c in the plots or in a table. It is somewhat confused the way you describe the that (a-c) figures are for mortality rates and (d-f) for species loss rate. You should mention, that capital letters represent significant effects based on the mixed model, right?. Do the other terms were not significant? And did you remove them? Or were they non-significant, but you leave them in the model.
It is necessary that authors include a table with all the terms and give the statistics to determine if were significant or not of the estimates of the model. With the information presented is not clear.

It is not clear if “year”, “Successional stage” and the other terms were non-significant?
How do the authors did know the p-value of “D”, “W”, “TxW”, etc?
lmer does not provide p-values. Were they obtained based with other package?
Authors definitively need to explain.

---

## Round 0.2 · accepted · Accept

Thank you for adding new references, the hypothesis is clear and the Introduction improved taking into account comments by the reviewers, as well as Results. The manuscript is ready for publishing.

Reviewer 1 ·

Basic reporting

I agree with the changes made to the document. The new manuscript is more robust and supports hypotheses, as well as greater clarity in all sections of the article. I consider that in its current form it is ready to be published in PeerJ

Experimental design

No comments

Validity of the findings

The findings presented here represent a good contribution to the state of the art in community ecology.

Reviewer 3 ·

Basic reporting

Authors have addressed or explained all my observations and comments on basic reporting from my previous review. I have no comments in the new version.

Experimental design

Authors attended all observations on experimental design and I believe the current version is now much more clear. I have no comments

Validity of the findings

Results are clear and, apart from that I was not able to see Figure S1, I have no comments.

Additional comments

Authors have made a great effort and have fairly addressed, not only all my comments, but those from the three reviewers.
I believe manuscript is ready to be published.